# Fermi polarons under strain-induced pseudomagnetic fields

Denis Yagodkin [1] ✉, Kenneth Burfeindt[1], Zakhar A. Iakovlev [2], Abhijeet M. Kumar[1], Adrián Dewambrechies[1], Oğuzhan Yücel[1], Bianca Höfer[1], Cornelius Gahl [1], Mikhail M. Glazov [2] & Kirill I. Bolotin [1] ✉

Excitons in Transition Metal Dichalcogenides (TMDs) acquire a spin-like quantum number, a pseudospin, originating from the crystal's discrete rotational symmetry. Here, we break this symmetry using a tunable uniaxial strain, effectively generating a pseudomagnetic field acting on exciton valley degree of freedom. Under this field, we demonstrate pseudospin analogs of spintronic phenomena such as the Zeeman effect and Larmor precession and determine fundamental timescales for pseudospin dynamics in TMDs. Finally, we uncover the bosonic – as opposed to fermionic – nature of many-body excitonic species using the pseudomagnetic equivalent of the $g$-factor spectroscopy. Our work is the first step toward establishing this spectroscopy as a universal method for probing correlated many-body states and realizing pseudospin analogs of spintronic devices.

Counterparts of magnetic phenomena arise in non-magnetic systems with two degenerate yet distinct states. A quantum number associated with this degeneracy can be treated as a spin analog, or "pseudospin", while the external perturbation lifting the degeneracy acts as a "pseudomagnetic field"[1–4]. Pseudomagnetic fields in systems ranging from photonic crystals[5,6] to inhomogeneously strained graphene[7,8] have been used to study topological phenomena, flat-band physics, and unconventional superconductivity[9–14]. In all these cases, the language of pseudomagnetic fields offers intuitive parallels to familiar magnetic phenomena, but applied to degrees of freedom that may remain unaffected by real magnetic fields.

One particularly appealing system for exploring pseudomagnetic phenomena is monolayers of Transition Metal Dichalcogenides (TMDs)[15–17]. There, a broken inversion symmetry gives rise to two energy-degenerate valleys at the K and K' points of the Brillouin zone that host tightly bound excitons (Fig. 1a). The pseudospin associated with this degeneracy can be initialized and read out optically: $\sigma^+$ ($\sigma^-$)-polarized light couples to excitons at the K (K') valleys (pseudospin up and down, respectively), whereas linear polarization couples to a coherent superposition of K and K' excitons, corresponding to an in-plane pseudospin[2,15–18]. The optical Stark effect has been used to lift the K/K' valley degeneracy,

effectively acting as a pseudomagnetic field[19,20]. Nevertheless, the high intensity of light pulses required to lift the degeneracy makes it challenging to study low-energy excitonic phenomena. In contrast, theory suggests that uniaxial mechanical strain produces a continuous, tunable in-plane pseudomagnetic field splitting states with dipole moments parallel ($X_b^0$) and orthogonal ($X_b^0$) to the strain axis (Fig. 1b)[1,2,17]. In this field, valley pseudospin is expected to exhibit analogs of magnetic Zeeman and Larmor effects. This raises a natural question: must pseudomagnetic phenomena always mirror those of real magnetic fields?

Unlike the conventional magnetic field, the strain-induced pseudomagnetic field preserves time-reversal symmetry and therefore affects only bosonic quasiparticles[21,22]. In contrast, the degeneracy of a fermionic Kramers pair cannot be lifted by a time-reversal-invariant perturbation. For example, neutral excitons ($X_0$), composite bosons formed by bound electron-hole pairs, are expected to split in a pseudomagnetic field (Fig. 1b). An intriguing situation occurs in doped TMDs when novel quasiparticles, charged excitons ($X^{+/-}$), arise. These quasiparticles can be described in two alternative ways, leading to different responses to the field (Fig. 1b). In the trion picture, they are fermionic three-particle states composed of a neutral exciton bound to a hole (electron)[15,23,24]. Such a state is

[1]Department of Physics and Halle-Berlin-Regensburg Cluster of Excellence CCE, Freie Universitat Berlin, Berlin, Germany. [2]Ioffe Institute, St. Petersburg, Russia. ✉e-mail: d.iagodkin@gmail.com; bolotin@zedat.fu-berlin.de

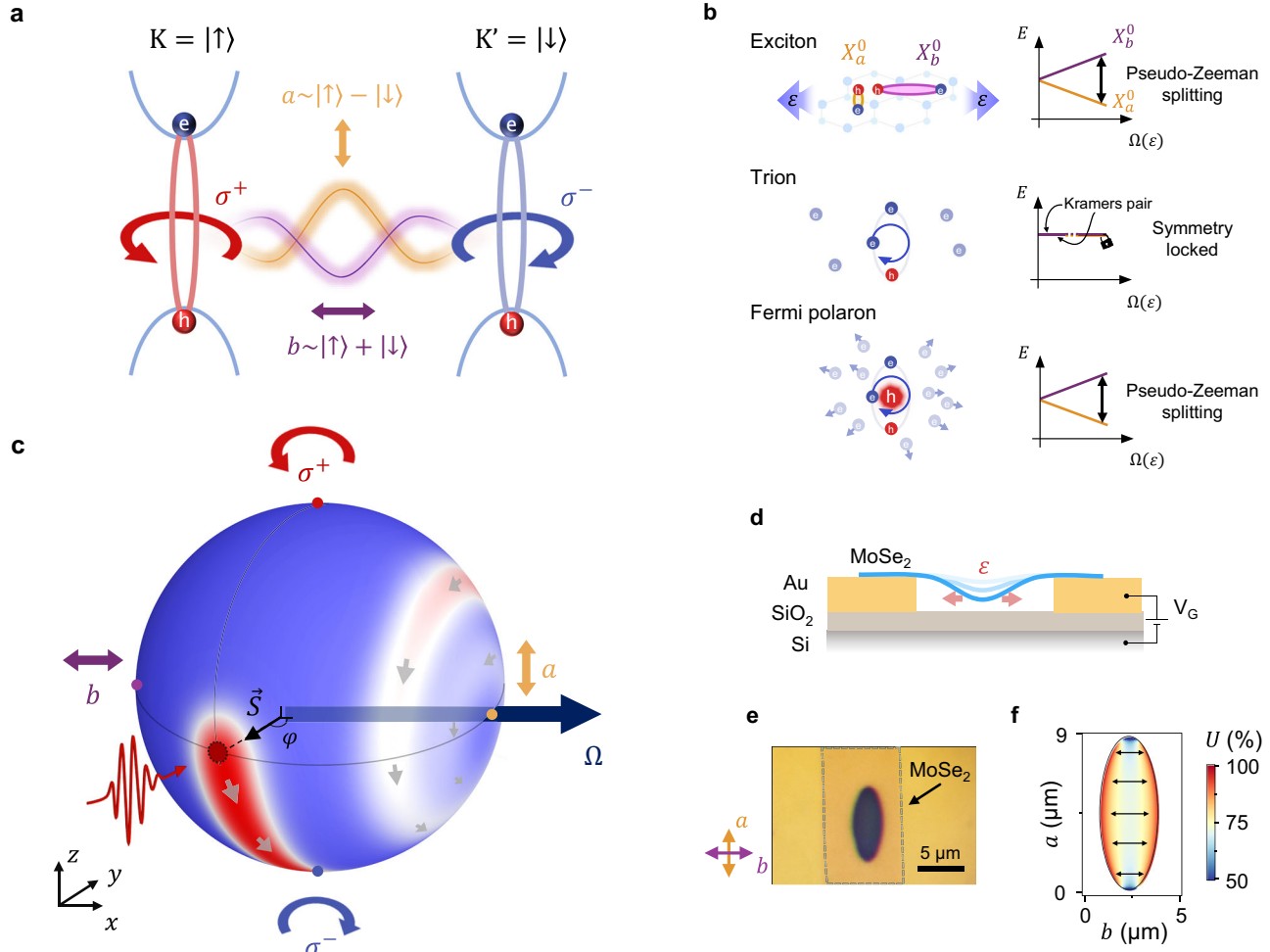

**Fig. 1 | Excitons, strain, and pseudomagnetic field. a** Different superpositions of excitons in K and K' valleys are excited by light with distinct polarizations. Circularly polarized light, $\sigma^+$ or $\sigma^-$, couples to K or K' excitons, respectively (red and blue arrows), whereas linearly polarized light (purple and orange arrows) generates superpositions of these excitons. **b** Uniaxial strain $\varepsilon$ produces an in-plane pseudomagnetic field $\mathbf{\Omega}(\varepsilon)$ that lifts the degeneracy of neutral excitons with dipole moments parallel and orthogonal to the straining axis. Under the same field, trions remain locked by time-reversal symmetry, while Fermi polarons split in energy, enabling pseudomagnetic $g$-factor spectroscopy. **c** Bloch sphere representation of pseudospin. Each coherent superposition of K and K' excitons corresponds to a pseudospin vector $\mathbf{S}$ on the Bloch sphere. The $\sigma^+$ or $\sigma^-$ circularly polarized light couples to the states at the poles, while linearly polarized light excites the states in the equatorial plane. In the presence of uniaxial strain, $\mathbf{S}$ undergoes damped Larmor-like precession around the strain-induced pseudomagnetic field $\mathbf{\Omega}$. **d** Straining technique: an applied gate voltage ($V_G$) induces tensile strain $\varepsilon$ (pink arrows) in suspended $MoSe_2$ or $WSe_2$ monolayer (blue) via electrostatic force. **e** Optical image of a suspended $MoSe_2$ monolayer. **f** COMSOL simulation of strain uniaxiality $U$ in a typical device.

Kramers protected and can only exhibit splitting in a real magnetic field. In the second "Fermi-polaron" (FP) or Suris tetron picture, a charged exciton is a neutral exciton correlated with the electron-hole pair inside the Fermi sea[24–29]. This state is a composite boson that can be split by a pseudomagnetic field[21,22]. Despite their different statistics, no measurement so far could conclusively distinguish between the Fermi-polaron and trion pictures.

Here, we resolve the debate about the nature of charged excitons using pseudomagnetic $g$-factor spectroscopy. To accomplish this, we introduce a method to generate a strong tunable strain-induced pseudomagnetic field in suspended monolayer TMDs at cryogenic temperatures. We take advantage of the time-independent nature, low disorder, and high magnitude of strain in TMDs to explore the effect of a pseudomagnetic field on various excitonic species. We first employ the pseudospin analogs of the Zeeman and Larmor effects to establish the strength of the pseudomagnetic field and obtain previously

unattainable material parameters. We then determine the symmetry of many-body excitonic states by measuring their pseudomagnetic $g$-factors. Our measurements show that both neutral and charged excitons can only be described as bosonic quasiparticles.

## Results

### Pseudospin in strained TMDs

The spatial symmetry of TMDs dictates that a linearly polarized photon in a state $\alpha|\sigma^+\rangle + \beta|\sigma^-\rangle$, with $|\alpha|^2 = |\beta|^2 = 1/2$, creates a coherent superposition of bright excitons with wavefunctions residing in K and K' valleys, $\Psi = \alpha|X_K\rangle + \beta|X_{K'}\rangle$. The spinor $\chi = (\alpha, \beta)$ then determines the pseudospin $\mathbf{S}$ in a similar way as the electron spin is defined in quantum mechanics: $\mathbf{S} = \left( \mathrm{Re}\,(\alpha\beta^*),\ \mathrm{Im}\,(\alpha^*\beta),\ |\alpha|^2 - |\beta|^2 \right)$. The application of mechanical strain breaks the underlying symmetries of TMDs, thereby affecting the pseudospin degree of freedom, see Supplementary Note S1[1,2]. The effect of strain on the exciton's pseudospin in the limit

of zero exciton momentum is described by the following Hamiltonian:

$$H = \begin{bmatrix} \frac{A}{2}\left(\varepsilon_{xx} + \varepsilon_{yy}\right) & \frac{B}{2}\left(\varepsilon_{xx} - \varepsilon_{yy} - 2i\varepsilon_{xy}\right) \\ \frac{B}{2}\left(\varepsilon_{xx} - \varepsilon_{yy} + 2i\varepsilon_{xy}\right) & \frac{A}{2}\left(\varepsilon_{xx} + \varepsilon_{yy}\right) \end{bmatrix}, \quad (1)$$

where $\varepsilon_{xx}$, $\varepsilon_{yy}$, $\varepsilon_{xy} = \varepsilon_{yx}$ are the components of the strain tensor, and $A$, $B$ are material-specific parameters. The diagonal terms describe the well-known energy shift of the excitons under biaxial strain at a rate $A \approx -100$ meV/%[30–32]. It is evident that K and K′ excitons, related by time-reversal symmetry, always remain energetically degenerate. However, the off-diagonal terms suggest that an application of uniaxial ($\varepsilon_{xx} \neq \varepsilon_{yy}$) or shear ($\varepsilon_{xy} \neq 0$) strain *mixes* excitons in K and K′ valleys. This effect becomes apparent if we rearrange the Hamiltonian in the form $H = H_0 + \frac{\hbar}{2}(\boldsymbol{\Omega} \cdot \boldsymbol{\sigma})$, where $H_0 = A\left(\varepsilon_{xx} + \varepsilon_{yy}\right)\sigma_0/2$ is the diagonal part of Eq. (1), $\boldsymbol{\Omega} = (B/\hbar)(\varepsilon_{xx} - \varepsilon_{yy}, 2\varepsilon_{xy}, 0)$, $\sigma_0$ is the identity matrix, and $\boldsymbol{\sigma} = (\sigma_x, \sigma_y, \sigma_z)$ is the vector of Pauli matrices acting in the pseudospin basis. This Hamiltonian is formally equivalent to that of a spin in a magnetic field, with the vector $\boldsymbol{\Omega}$ playing the role of the pseudomagnetic field. We therefore expect the presence of analogs of magnetic phenomena in strained devices.

## Generation of pseudomagnetic field and detection of a pseudospin

We induce a strong pseudomagnetic field at cryogenic temperatures using a technique based on tensioning of a suspended monolayer with electrostatic force (Fig. 1d) that we recently developed[30]. Our approach overcomes the limitations of previous methods that function only at elevated temperatures, leaving pseudomagnetic phenomena largely unexplored[33,34]. Moreover, our clean samples ensure a long lifetime and low decoherence rate of excitons. We focus on two materials representative of the TMDs family: monolayer MoSe$_2$, chosen for its well-understood and rather simple excitonic spectrum[35], and WSe$_2$, selected for its long coherence time of excitons comparable to their lifetime[36–39].

Our device consists of a TMD monolayer suspended over a trench in an Au/SiO$_2$/Si stack (Fig. 1d, e). A gate voltage, $V_G$, applied between the Si substrate and the sample induces an electrostatic pressure and strains the TMD, with the strain distribution defined by the trench geometry (see Note S2 for the calibration of applied strain). For an elliptical trench with major axis $a$ and minor axis $b$ ($a \gg b$), a predominantly uniaxial strain is induced along $b$, which we quantify via the degree of uniaxiality, $U = (\varepsilon_{bb} - \varepsilon_{aa})/(\varepsilon_{bb} + \varepsilon_{aa})$. Specifically, we use an

ellipse with $a = 8\,\mu m$ and $b = 3\,\mu m$, which ensures high uniaxiality $U \approx 80\%$ (Fig. 1f), while maintaining strain uniformity $\frac{\Delta\varepsilon}{\varepsilon} < 10\%$ within the laser spot of ~1 μm (Fig. S1a–c). Conversely, a device with a circular trench experiences uniform biaxial strain ($U \approx 0$) in the center of the membrane (Supplementary Fig. S1e–g).

In a prototypical experiment, the uniaxial strain generates a pseudomagnetic field, $\boldsymbol{\Omega}$, along the $x$-axis in pseudospin space (Fig. 1c). In analogy to the Zeeman effect, we expect the exciton energy to depend on the orientation of its pseudospin $\boldsymbol{S}$ with respect to $\boldsymbol{\Omega}$, being minimal when the two vectors are aligned. To study this effect, we use the fact that the pseudospin orientation on the Bloch sphere determines the polarization of a photon coupled to this pseudospin. Specifically, we access the energy of the states with pseudospin along the equator of the Bloch sphere by recording the linear polarization-resolved photoluminescence (PL) spectra.

In analogy to the Larmor effect, the pseudospin along the $y$-axis in pseudospin space − that is, excited by light polarized along a direction at 45° with respect to the strain axis − undergoes damped precession around $\boldsymbol{\Omega}$ (red cloud in Fig. 1c). Such precession is signaled by the appearance of the pseudospin component $S_z$, while the damped nature of the precession leads to the development of a pseudospin component aligned with the field, $S_\parallel$. We experimentally determine the components of pseudospin from polarization-resolved PL spectra as $S_z = \frac{I(\sigma^+) - I(\sigma^-)}{I(\sigma^+) + I(\sigma^-)}$ and $S_\parallel = \frac{I(a) - I(b)}{I(a) + I(b)}$, where $I(\sigma^+)$ and $I(\sigma^-)$ are the intensities of $\sigma^+$ or $\sigma^-$ polarized light; $I(a)$ and $I(b)$ are intensities polarized along and perpendicular to the strain axis, respectively[40].

We begin by studying an analog of the Zeeman effect to characterize the achievable field strength. Subsequently, we investigate the Larmor effect in this field. The characteristic time scales extracted from these measurements provide insights into the mechanisms of pseudospin polarization loss and strategies to suppress it. We finally develop a counterpart of $g$-factor measurements to uncover the nature of many-body states.

## Zeeman splitting in pseudomagnetic field

Figure 2 a shows the polarization-resolved PL spectra of X$^0$ emission energy in an unstrained MoSe$_2$ ("Methods"). The orange and purple spectra, corresponding to the polarization along the major ($a$) and minor ($b$) axes, respectively, show the expected nearly identical emission energy, $E_a = E_b$. However, a relative energy shift emerges when uniaxial strain is applied ($\varepsilon = \varepsilon_{bb} - \varepsilon_{aa} = 0.4\%$; Fig. 2b). Indeed, a false-color map of the polarization-resolved PL spectra of the strained

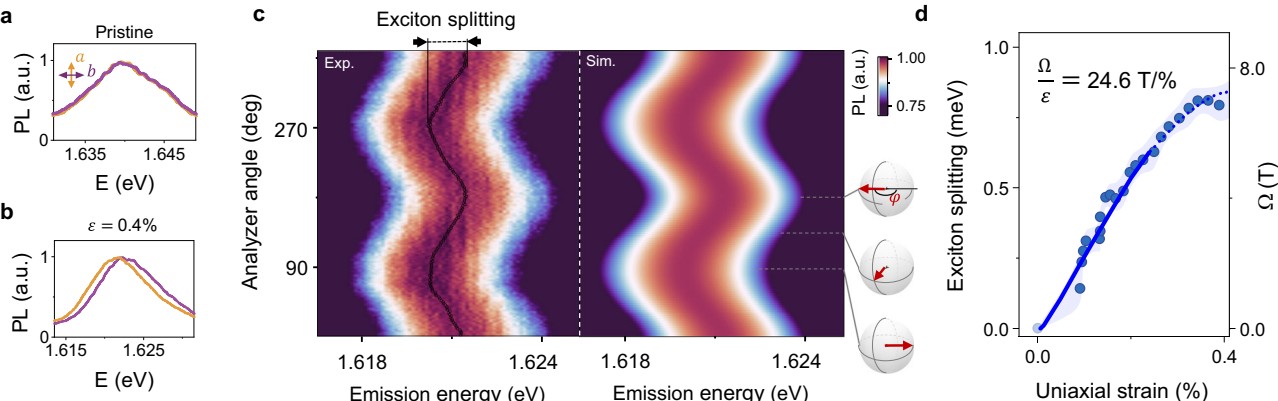

**Fig. 2 | Pseudo-Zeeman effect. a, b** Polarization-resolved PL spectra at near-zero strain (top panel) and under 0.4% uniaxial strain (bottom panel) in the region of neutral exciton X$^0$ in MoSe$_2$. The emission energy of X$^0$ becomes polarization-dependent under strain, with higher energy along the direction of uniaxial strain $b$ (purple) than orthogonal to it (orange). Polarizations of both excitation and detection are linear and co-polarized. **c** Normalized PL spectra for the same device as a function of the analyzer angle at 0.4% strain, along with the simulations (circles mark the extracted peak position). Note, that the angle $\varphi$ between the probed pseudospin $\boldsymbol{S}$ and $\boldsymbol{\Omega}$ is twice the angle between the polarizer (analyzer) axis and the strain direction $b$ (side panel). **d** The energy splitting between the excitons with pseudospin aligned along or opposite to the pseudomagnetic field, interpreted as pseudo-Zeeman splitting, extracted from (**c**). The shaded area represents the uncertainty.

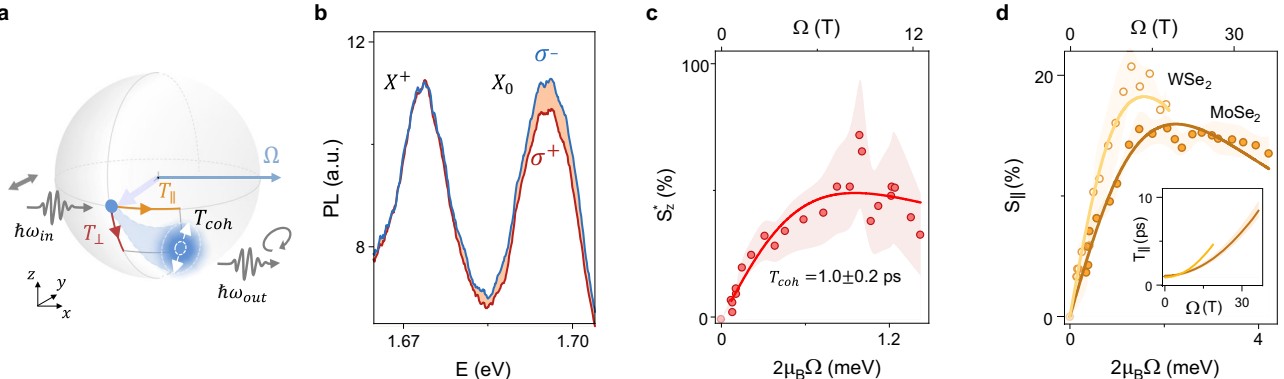

**Fig. 3 | Pseudo-Larmor effect. a** Schematics of the expected Larmor-like dynamics of pseudospin in a pseudomagnetic field. **b** Circular-polarization-resolved PL spectra of WSe$_2$ under 8 T (0.9 meV) pseudomagnetic field, excited by linearly polarized light. The rotation of the exciton's pseudospin is manifested as an asymmetry between $\sigma^-$ and $\sigma^+$ emission of the neutral exciton (X$^0$). **c** The $S_z^*$ component of the pseudospin vs. the pseudomagnetic field strength in WSe$_2$ (red points) and fit to the model Eq. (2) (red line), top and bottom x-axes are the pseudomagnetic field strength and the corresponding excitonic splitting, respectively. The shadow represents uncertainty. **d** The component of the pseudospin along the field, $S_{\parallel}$, vs. field strength in MoSe$_2$ and WSe$_2$ and fit to our theoretical model Eq. (2). Inset: the dependence of $T_{\parallel}$ on the pseudomagnetic field strength in MoSe$_2$ and WSe$_2$ (dark and bright orange lines, respectively).

sample (left panel in Fig. 2c) reveals a clear sinusoidal dependence of the X$^0$ emission energy on the detection polarization direction. The minimum and maximum of the X$^0$ emission energy correspond to $\boldsymbol{S}$ oriented along and opposite to $\boldsymbol{\Omega}$, respectively (see schematic in Fig. 2c). This strain-induced energy splitting between the two orthogonal polarization directions is, in fact, analogous to the Zeeman effect for pseudospins; hence, we term it pseudo-Zeeman splitting.

To quantify the established pseudo-Zeeman effect, we fit the data in Fig. 2c using $E(\varphi) = E_0 + (\hbar\Omega/2)\cos\varphi$, where the term $E_0 = A(\varepsilon_{xx} + \varepsilon_{yy})/2$ describes the strain-induced redshift in X$^0$ energy compared to the unstrained state (see Eq. 1) and $\varphi$ is the angle between the exciton pseudospin and pseudomagnetic field. The extracted pseudomagnetic field grows linearly at small strain level (< 0.4%) at a rate of $B = 24.6 \pm 2.5$ T/% in MoSe$_2$ (solid line in Fig. 2d) and $16.1 \pm 1.8$ T/% in WSe$_2$ (Supplementary Fig. S2) corresponding to $2.9 \pm 0.3$ meV and $1.9 \pm 0.2$ meV, respectively. Following an established convention[7,41,42], we used the free-electron gyromagnetic g-factor $g = 2$ (corresponding to $2\mu_B = 0.116$ meV/T, with $\mu_B$ being the Bohr magneton) to convert the measured splitting into an equivalent pseudomagnetic field in Tesla solely for easier comparison with conventional magnetic effects. To emphasize the difference between pseudomagnetic and real magnetic fields, we also provide the exciton splitting corresponding to the field in units of energy, whenever appropriate. At higher strain level, the apparent dependence of exciton splitting becomes sublinear (Supplementary Fig. S3), which we attribute to a reduced intensity of the higher energy pseudo-Zeeman-split state when the energy separation exceeds the thermal energy ($k_B T \approx 1$ meV). The model based on this mechanism closely aligns with the observed behavior of X$^0$ (simulation in Fig. 2c, Supplementary Note S7) and the extracted splitting (dotted line in Fig. 2d). In addition, the splitting is close to the expected value in the optical reflectivity measurements (Supplementary Fig. S4). Therefore, in the following, we assume a linear dependence of $\Omega$ on strain, with $\Omega$ reaching $43 \pm 6$ T ($5.0 \pm 0.7$ meV) in MoSe$_2$ at our highest applied strain of 1.6% (Fig. S3). Finally, we note that the pseudo-Zeeman effect is absent in biaxially strained devices ($\Omega = 0$), an experimental situation realized in circular trenches (Supplementary Fig. S5). This finding further confirms that the observed behavior in Fig. 2 results from the pseudospin Zeeman effect and rules out artifacts related to, e.g., spurious plasmonic effects, biaxial strain, etc.

### Strain control of pseudospin dynamics

Our next objective is to gain control over pseudospin dynamics; to this end, we explore the pseudospin analog of Larmor precession and

quantify the characteristic pseudospin relaxation times. A hallmark of Larmor precession is the emergence of circularly polarized PL emission under linearly polarized excitation (Fig. 3a). Figure 3b shows circular polarization-resolved PL spectra of WSe$_2$ at $\Omega = 8$ T (0.9 meV) corresponding to $\varepsilon = 0.5\%$. Under the strain-induced pseudomagnetic field, a prominent asymmetry between the $I(\sigma^+)$ and $I(\sigma^-)$ intensities at the X$^0$ emission energy (red and blue, respectively) emerges, whose sign depends on the excitation polarization direction (Supplementary Figs. S6 and S7). This observation is striking, as a circularly polarized emission under linear excitation can only be caused by the breaking of either time-reversal or spatial symmetries. Since the magnetic field is absent in our experiments and the asymmetry is detected only when a pseudomagnetic field is induced (Supplementary Fig. S8), we conclude that the pseudomagnetic field alone is responsible for the observed Larmor-like effect.

To gain insight into the mechanism of pseudospin dynamics and relaxation, we develop a theory of pseudo-Larmor precession. The full model is provided in Supplementary Note S1, we illustrate the concept here with an example based on the Bloch equation for population-averaged pseudospin dynamics

$$\frac{\partial \boldsymbol{S}}{\partial t} + \frac{\boldsymbol{S}}{\tau} + \boldsymbol{S}_{\perp} \times \boldsymbol{\Omega} + \frac{\boldsymbol{S}_{\perp}}{T_{coh}} + \frac{\boldsymbol{S}_{\parallel} - \boldsymbol{S}_0}{T_{\parallel}} = \boldsymbol{G}, \quad (2)$$

where $\boldsymbol{G}$ is the pseudospin generation rate defined by the excitation intensity and polarization, $\boldsymbol{S}_0$ describes the quasi-equilibrium (thermal) pseudospin induced by the pseudomagnetic field (Fig. 3a). The characteristic times are: exciton lifetime ($\tau \approx 2$ ps)[37,38,43-48], period of Larmor precession ($T_{\perp} = 2\pi/\Omega$), $T_{coh}$ is the coherence time that determines relaxation of the pseudospin components transverse to the field, and $T_{\parallel}$ characterizes the time over which thermal equilibrium between the split sublevels is established (for the relation of Eq. (2) to the microscopic model, see Supplementary Notes S1, S3, and S4). The microscopic model accounts for the exciton longitudinal-transverse splitting caused by the electron-hole exchange interaction. This splitting induces an effective wavevector-dependent pseudomagnetic field $\Omega^{LT}$, which is present even in an unstrained monolayer and leads to the loss of pseudospin coherence by the Dyakonov-Perel mechanism[18,49]. A strain-induced pseudomagnetic field suppresses $\Omega^{LT}$-induced depolarization, which significantly increases both $T_{coh}$ and $T_{\parallel}$ (Supplementary Note S3). Our goal is to experimentally determine these two timescales that define pseudospin dynamics yet remain unknown.

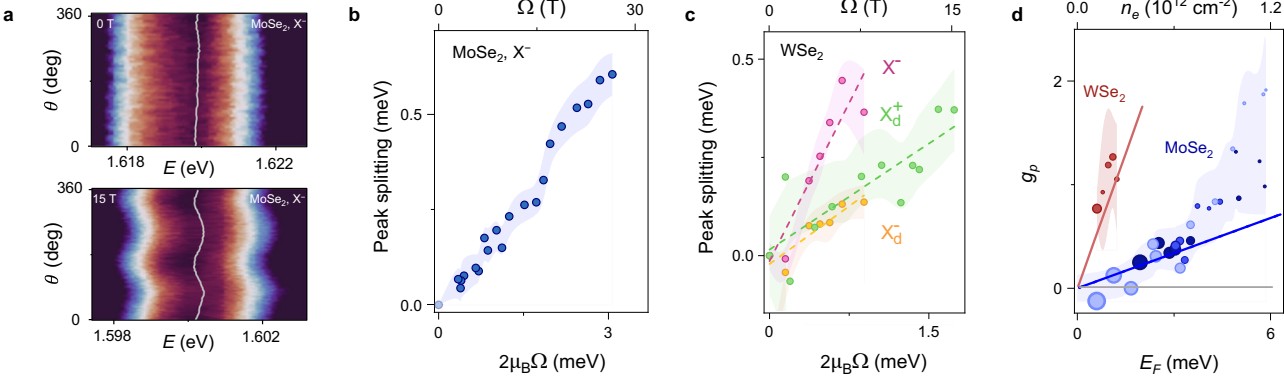

**Fig. 4 | Charged excitons under pseudomagnetic field. a** False-color map of polarization-resolved PL of the charged exciton ($X^-$) in monolayer MoSe$_2$. Under a strain-induced pseudomagnetic field, a prominent pseudo Zeeman splitting appears. **b** Splitting of the negatively charged exciton as a function of pseudomagnetic field strength in doped MoSe$_2$. The observed splitting is consistent with the polaronic character of the charged exciton. **c** Peak splitting of bright ($X^-$) and dark ($X_d^+$, $X_d^-$) charged excitons in WSe$_2$ as a function of pseudomagnetic field strength. **d** The dependence of pseudospin $g$-factor $g_p$ of bright FP on Fermi energy in WSe$_2$ (red points) and MoSe$_2$ (blue points), alongside theoretical predictions[21] (red and blue solid lines, respectively). The size of each point is proportional to strain, and color shades mark different experimental runs with different initial carrier densities.

In a simple case of unitary excitation along the $y$ pseudospin axis, $G\tau_\perp = (0, 1, 0)$, the steady-state solution of Eq. (2) is $S_z = \tau_\perp \Omega / \left[1 + (\tau_\perp \Omega)^2\right]$, where $1/\tau_\perp = 1/T_{coh} + 1/\tau$, note that $\Omega$ in this equation has units of rad/s (Supplementary Note S1). Intuitively, ensemble averaged $S_z$ probed by PL grows linearly with $\Omega$ when the average rotation angle for pseudospins during their lifetime is small, $\tau_\perp \Omega \ll 1$. At higher field strengths, the pseudospin undergoes multiple rotations around the Bloch sphere during the exciton lifetime, reducing the average pseudospin polarization similar to the Hanle effect in real magnetic fields. To experimentally realize the scenario of unitary excitation, we consider the reduced pseudospin $S_z^*(\Omega)$, normalized to the measured generation rate at the corresponding field $G(\Omega)$ (Supplementary Note S3).

Figure 3c shows the experimentally obtained dependence of $S_z^*$ on the pseudomagnetic field in WSe$_2$, along with a fit using the solution of Eq. (2). This fit yields $T_{coh} = \tau_\perp \tau / (\tau - \tau_\perp) = 1.0 \pm 0.2$ ps in the regime of high field strength, which is longer than the coherence time measured in the unstrained samples ($T_{coh} \sim 0.5$ ps[37,38]) due to the influence of the pseudomagnetic field (Supplementary Note S3). Finally, the large pseudospin polarization, $S_z = 50\%$, demonstrates the strong potential of the pseudomagnetic field for manipulating the exciton pseudospin.

To determine $T_\parallel$, we examine Eq. (2) under unpolarized excitation conditions, which are experimentally realized at high detuning of the excitation energy from the $X^0$ resonance so that all induced polarization is lost. In this case, $G \rightarrow 0$ and only field-induced $S$ appears in the form $S_\parallel = \tau / (\tau + T_\parallel) \times \tanh\left[\hbar\Omega/(2k_BT)\right]$ (Supplementary Note S1).

This expression suggests that the initially unpolarized pseudospins tend to align along $\Omega$, acquiring a pseudospin polarization within a thermal distribution. The induced polarization saturates when the pseudo-Zeeman splitting exceeds the thermal energy ($k_BT \approx 1$ meV), with its maximum value determined by the ratio of the relaxation time $T_\parallel$ to the lifetime $\tau$.

The experimentally observed $S_\parallel$ vs. $\Omega$ (Fig. 3d) matches these expectations. At low field strengths ($\Omega < 10$ T (1.2 meV)), we observe a linear increase in $S_\parallel$. At higher fields, the polarization reaches the expected plateau, $S_\parallel(\hbar\Omega \gg k_BT) = \tau / (\tau + T_\parallel)$. From the value of $S_\parallel \approx 20\%$ at the plateau in both MoSe$_2$ and WSe$_2$, we find the pseudospin relaxation time $T_\parallel \sim 10$ ps (Supplementary Note S3), significantly longer than the exciton coherence $T_{coh} \sim 0.5$ ps and lifetime $\tau \approx 2$ ps in these samples[50,51]. This slowdown of the relaxation time arises because the pseudomagnetic field suppresses pseudospin decay dominated by $\Omega^{LT}$ (see Supplementary Note S3). Using a model that accounts for this effect (Supplementary Note S1), we fit $S_\parallel$ and find that the relaxation time increases from 1 to 8 ps over the studied range of field strengths (inset in Fig. 3d). Furthermore, this analysis allows us to extract the field responsible for loss of pseudospin coherence, yielding the root-mean-square values $\Omega_{WSe_2}^{LT} = 10.4 \pm 1.3$ T (1.2 meV) in WSe$_2$ and $\Omega_{MoSe_2}^{LT} = 12.0 \pm 1.1$ T (1.4 meV) in MoSe$_2$ in reasonable agreement with the model predictions (Supplementary Note S4). To the best of our knowledge, this constitutes the first measurement of this fundamental parameter.

## Many-body states under pseudomagnetic field

Our ultimate goal is to investigate complex many-body states beyond neutral excitons under the pseudomagnetic field and to showcase the unique capacity of our technique to reveal their intrinsic structure. Two critical aspects remain experimentally unexplored. First, recent theoretical studies have suggested that trions and FPs show contrasting behaviors under a pseudomagnetic field due to the distinct response to time-reversal symmetry[21,22]. That suggests a possibility of a $g$-factor-like measurement to distinguish the two descriptions of charged excitons. We define the pseudomagnetic $g$-factor ($g_p$) as $\Delta E = \frac{g_p}{2} \hbar\Omega$, where $\Delta E$ is pseudo-Zeeman splitting, with $g_p = 0$ signifying a trion and $g_p \neq 0$ indicating a Fermi polaron nature of the charged exciton. Second, since the nature of trions and FPs are strongly affected by the density of charge carriers (Fig. 1b), the magnitude of $g_p$ is expected to depend on the Fermi energy ($E_F$). Specifically, $g_p$ can be expressed as $g_p(E_F) = 2\Delta E_{FP}(E_F)/\Delta E_X$, where $\Delta E_X = \hbar\Omega$, and $\Delta E_{FP} = \frac{g_p(E_F)}{2} \hbar\Omega$. In our devices, an applied gate voltage varies the Fermi energy together with strain, enabling measurement of the pseudomagnetic g-factor.

To test these predictions, we probed the response of charged excitons in MoSe$_2$ and WSe$_2$ under an applied pseudomagnetic field (Fig. 4a–c). We used the same experimental configuration and analysis as in the study of the pseudo-Zeeman effect of neutral excitons. Figure 4a shows the pseudomagnetic-field-induced energy splitting of the negatively charged excitons ($X^-$) in doped MoSe$_2$ ($n_e > 1 \times 10^{12}$ cm$^{-2}$) with pseudospins aligned along and opposite to the pseudomagnetic field. The observed finite energy splitting for $X^-$ is similar to what was seen previously for neutral excitons (Fig. 2d), although with a much lower magnitude (Fig. 4b). The observation of pseudo-Zeeman splitting of the $X^-$ state provides conclusive evidence of their Fermi polaron nature and establishes their bosonic statistics.

In contrast to $MoSe_2$, $WSe_2$ hosts a plethora of additional many-body states (Supplementary Fig. S2), including positively and negatively charged bright excitons ($X^+$ and $X^-$), neutral and charged dark excitons ($X_d$, $X_d^+$, and $X_d^-$), biexcitons (XX), and phonon replicas ($X_p$)[52,53]. We observe a considerable strain-dependent energy splitting of $X^-$, $X_d^+$, and $X_d^-$ in that material (Fig. 4c), which confirms their Fermi polaronic nature. The dark species demonstrate lower splitting and an overall lower pseudomagnetic $g$-factor, $g_p(X_d^{+/-}) \approx 0.8$, compared to the bright ones, $g_p(X^-) \approx 2.0$ for the same doping level. We note that the low intensity of biexcitons and phonon replicas prevents us from extracting their splitting, while $X^+$ is only visible at low pseudomagnetic fields (Supplementary Fig. S2).

Finally, we use the pseudomagnetic $g$-factor to explore the effect of Fermi energy (charge density) on the character of charged excitons. The pseudomagnetic $g$-factor of FPs vs. Fermi energy is plotted in Fig. 4d; the size of each point is proportional to the uniaxial strain (see Supplementary Note S5 for Fermi energy estimation). We find that for low Fermi energy, $g_p$ is nearly zero despite a large pseudomagnetic field, which is consistent with the convergence of Fermi polaronic and trionic pictures in this regime. Meanwhile, at a larger $E_F$, the splitting of the charged exciton approaches that of a neutral exciton. This behavior is expected, as the attractive Fermi polaron splitting inherits the neutral exciton splitting and saturates at this value. Indeed, theory predicts[21] that the attractive polaron $g$-factor depends linearly on Fermi energy $E_F$ (Supplementary Note S6). Moreover, the predicted value of $g_p$ for charged excitons in $WSe_2$ (red line in Fig. 4d) is higher than that in $MoSe_2$ (blue line in Fig. 4d) for the same doping level, due to the mixing of the intervalley and intravalley states[21]. A close match between the experimental results and theoretical predictions further supports the tuning of FP character by induced charge density. Overall, our results establish the pseudo-Zeeman splitting as a tool to assess the symmetry and statistics of excitonic states.

## Discussion

Our technique to study and manipulate pseudospin opens multiple new possibilities. First, the interplay between magnetic and pseudomagnetic fields in the same device is promising to reveal unique effects[54]. The presence of strongly coupled spin and valley pseudospin degrees of freedom with distinctive timescales should cause complex and hitherto unstudied dynamics. Second, our results indicate a rotation of the pseudospin during pseudo-Larmor precession. The pseudospin dynamics can be probed in the time domain by observing an oscillating signal in, e.g., time-resolved Kerr rotation microscopy[51,55,56]. Third, the coupling between the pseudospin and momentum can lead to the pseudomagnetic counterparts of spin-orbit phenomena such as the anomalous Hall, quantum spin Hall, and Rashba-like effects[54,57–60]. The complex nature of momentum/pseudospin coupling should significantly alter these effects compared to their classical counterparts[9,10,61]. Finally, the effects studied above suggest several potential applications. For example, the Larmor precession of pseudospin should generate THz emission with the frequency controlled by the amount of strain, potentially enabling a broadly tunable THz emission source[62,63]. If the coherence time could be extended, e.g., in TMD heterostructures[48,64,65], pseudospin-based devices could be considered as qubits potentially suitable for the effective transduction of mechanical and optical information.

## Methods

### Sample fabrication

The devices were fabricated by dry transfer of mechanically exfoliated TMD flakes onto elliptical ($8 \times 3\,\mu m$) or circular trenches (diameter ~ $5\,\mu m$), which were wet-etched via hydrofluoric (HF) acid in an Au/Cr/$SiO_2$/Si stack[30,31]. The strain in the membrane was induced by applying a gate voltage (typically up to $\pm 210\,V$) between the TMD flake (electrically grounded) and the Si back gate of the chip. The strain in the center was characterized using laser interferometry (see Supplementary Note S2).

### Optical measurements

The devices were measured inside a cryostat (CryoVac Konti Micro) at a base temperature of 10 K. Photoluminescence (PL) measurements were carried out using a Kymera 193i spectrograph and continuous-wave (CW) lasers with either $\lambda = 685\,nm$ ($8\,\mu W$) for quasi-resonant excitation or $\lambda = 532\,nm$ ($6\,\mu W$) for detuned excitation. The lasers were tightly focused at the center of the membrane with a spot diameter of approximately $0.8\,\mu m$. The excitation polarization was controlled using a half-wave plate (RAC 4.2.10, B. Halle) placed before the objective (Olympus LMPlan 50x, 0.5 NA) to reduce polarization loss. The detection polarization was set using a combination of either a half-wave plate or a quarter-wave plate (for linear and circular detection, respectively) and an analyzer (GL 10, Thorlabs) before the spectrometer. To minimize the influence of coherent effects on pseudo-Zeeman splitting, we maintained excitation and detection co-polarized. The Fermi polaron splitting was measured in a Cryostation s100 cryostat (Montana Instruments) with an Isoplane 320 spectrometer (Teledyne Princeton Instruments), using a 532 nm CW laser focused to a diffraction-limited spot with an objective (Zeiss Epiplan 100x, 0.75 NA).

## Data availability

Data supporting the findings of this study are available on Zenodo (DOI: 10.5281/zenodo.14844313). Additional data can be provided by the corresponding author upon request.

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

## Acknowledgements

The Berlin groups acknowledge the Deutsche Forschungsgemeinschaft (DFG) for financial support through the Collaborative Research Center TRR 227 Ultrafast Spin Dynamics

(project B08), Project GZ: BO 5142/4-1, the Priority Program SPP 2244, the German Excellence Strategy - EXC3112/1 - 533767171 (Center for Chiral Electronics), and the Federal Ministry of Education and Research (BMBF, project 05K22KE3). The Saint Petersburg group acknowledges financial support by the RSF Project 23-12-00142 (theory); Z.A.I. gratefully acknowledges the BASIS foundation. K.I.B. acknowledges illuminating discussions with Christiane Koch, Robert Bittl, and Stephanie Reich.

## Author contributions

D.Y. and K.I.B. conceived the project. Z.A.I. and M.M.G. developed the theory. D.Y., A.M.K., A.D., and C.G. designed the experimental setup. D.Y., K.B., A.M.K., and B.H. prepared the samples. D.Y., K.B., A.M.K., and A.D. performed the optical measurements. O.Y. performed mechanical simulations. D.Y. and K.B. analyzed the data. D.Y. and K.I.B. wrote the manuscript with input from all co-authors.

## Funding

## Competing interests

The authors declare no competing interests.
