## [Transparent Peer Review file · Nature Communications]

Fermi polarons under strain-induced pseudomagnetic fields

Corresponding Author: Professor Kirill Bolotin

Version 0:

Reviewer comments:

Reviewer #1

(Remarks to the Author)

The manuscript entitled “Fermi polarons under large pseudomagnetic fields” by Denis Yagodkin et al. report observation on luminescence from MoSe₂ and WSe₂ monolayers under strain. The authors claim that pseudospin of excitonic quasiparticles can be manipulated by tunable uniaxial strain which acts like pseudomagnetic fields. Although the claimed observation poses an interesting perspective, I have several concerns and questions on their analysis. Therefore, I cannot recommend the publication in Nature Communications.

1. They claim that they can generate pseudomagnetic fields exceeding 40 Tesla. However, it is not clear how this pseudomagnetic field is estimated from observed energy shifts. To quantify pseudomagnetic fields, one would need g-factors of excitonic quasiparticles. In their manuscript, they seemed to use $2\mu_B = 0.116 \text{ meV/T}$ which is attributed to electron's gyromagnetic factor. How does the author justify this g-factor for their observation for neutral excitons and fermi polaron? One should carefully consider g-factors for electrons and holes which can contribute to form excitonic quasiparticles.
2. They show pseudo-Zeeman effect for MoSe₂ monolayers while they show pseudo-Larmor effect for WSe₂ monolayers. Why can they show both effects for the same materials? According to their claim, the pseudo-Zeeman effect is the driving mechanism or pseudo-Larmor effect. For the coherence of the overall discussion, it is much better to choose one material to demonstrate both effects.
3. They claim that pseudospin dynamics can be controlled by strains. They define S_z and $S_{//}$ to describe quantum coherent states of pseudospin in Bloch sphere as shown in their Fig.1c. Under linearly polarized laser excitation, they observe photoluminescence signals that shows stronger circular polarization component for σ_- than σ_+ . They interpretate their observation as a result of pseudo-Larmor effect in Fig.3. However, in order to claim such effects, the authors should systematically show orientation dependence of their linearly polarized excitation with respect to the pseudomagnetic field direction. When their pseudospin direction is along the pseudomagnetic field direction, they should not observe the circular polarization dependence in their PL.
4. There are already several works on controlling pseudospin in TMD monolayers using pseudomagnetic fields or real magnetic fields. For example, Nature Materials 14, 290–294 (2015), Science 346, 1205 (2014), Nature Physics 13 (1), 26-29 (2017), and several. They should show how their works provide new novel findings by comparing to the previous works.

Reviewer #2

(Remarks to the Author)

The manuscript titled ‘Fermi polarons under large pseudomagnetic fields’ by Denis Yagodkin et al describes the manipulation of valley pseudospin of neutral excitons in monolayer MoSe₂ and WSe₂ using an uniaxial strain. The authors claim an in-plane pseudomagnetic over 30 T and an enhanced intervalley coherence time which allows up to three full rotations along the polar direction in the Bloch sphere. In the last part, the authors also conclude the nature of the charged exciton, after comparing its pseudo-Zeeman splitting with theoretical predictions. Although the physics discussed in the work is interesting and the theoretical analysis is very thorough, I feel more experimental evidence is needed to support the authors' claims, as explained in details below.

1. Although the authors claim to have generated a pseudomagnetic field up to 30 T, they only experimentally demonstrate an exciton splitting of 0.7 meV through the polarization dependence of the photoluminescence spectra, which corresponds to a 7-T field. The claimed large field is an extrapolation from the low field limit based on the calibrated strain. I understand the authors' argument that when the splitting becomes larger, the high energy state is not thermally populated, so the splitting is no longer reflected in PL. But relying on pure theoretical modelling is risky as it contains some assumptions that may not apply to all experimental conditions. For example, under the largest strain condition, the extrapolated exciton splitting becomes comparable to the thermal energy, so the Bose-Einstein distribution should be used, instead of Boltzmann distribution. In the meantime, the uniaxial strain calibration also relies heavily on the finite element analysis in the linear regime, while we know the mechanical problem can become highly nonlinear when the strain is large. Therefore, I would suggest the authors perform some additional experiments to verify some of the key conclusions of their models. For example, I imagine the exciton splitting under large strain can be directly measured in the absorption spectrum through polarization resolved reflection contrast experiments.

2. If the pseudospin does rotate in the vertical plane for three rounds before it becomes incoherent, I expect the magnetic field dependence of the integration S_z^* signal should exhibit some damped oscillation, which is currently missing in Fig. 3c or S2. Actually, there is currently no data for S_z^* for pseudomagnetic field larger than 12T despite of the claim...

3. The authors claim the T_{parallel} is enhanced by about ten times and the T_{coh} is doubled as the pseudomagnetic field increases, but I am confused about the experimental evidence. In Fig. S8, the authors made a comparison plot between two theoretical models, one assuming a constant T and the other assuming a field-dependent T . First, in Fig. S8b, my understanding is the data quality cannot distinguish two models. Second, in Fig. S8a, there exists some difference at the large field limit, but the authors' assumption is actually the opposite - the two models should give the same T_{parallel} at the large field limit and different T_{parallel} at the small field limit, where both models somehow fit the data well. Therefore, I do not find sufficient experimental evidence to support the authors' claim.

4. How do the authors perform the doping dependence study of g_p as in Fig. 4d is very unclear to me. For MoSe₂, it seems the sample is initially chemically doped so when no gate voltage is applied, the Fermi level is highest (6 meV) and the strain is minimal. As the gate voltage is applied, the doping becomes smaller and the strain increases. The exciton splitting also increases with pseudomagnetic field, so the uncertainty in g_p decreases as E_f decreases. But if this is the case, why are there some data points near 4 meV which shows negligible strain and significant electrostatic doping? Can the authors provide some doping dependent spectra as in Fig. S2 d? On the other hand, in the WSe₂ sample, why are there some low-strain data points at the end and in the middle of the data series? It doesn't seem consistent with Fig. S2 d either, which suggests the sample is initially intrinsic.

5. A technical suggestion - since the excitation condition is critical to the coherent part of PL, the authors may want to clarify that for Fig. 2.

Reviewer #3

(Remarks to the Author)

Version 1:

Reviewer comments:

Reviewer #1

(Remarks to the Author)

Denis Yagodkin et al. have improved their discussion of the observed phenomena in the revised manuscript. However, I remain concerned about several critical issues that directly affect the validity of their main claims:

1. Definition of the in-plane pseudomagnetic field.

The magnitude of the pseudomagnetic field in their work is essentially a mathematical parameter without a clear connection to any physical property. In their rebuttal, the authors themselves acknowledged the difficulty of physically defining the in-plane excitonic g-factor. As they also cite (Nat. Commun. 13, 499), the in-plane excitonic g-factor is known to be effectively zero. Despite this, they simply apply the free-electron g-factor to excitonic quasiparticles. This approach is inappropriate, since the g-factor of an exciton is determined jointly by both electron and hole contributions.

The authors write in their revised manuscript:

“We used the free-electron gyromagnetic factor $g = 2$ solely to convert the measured splitting into an equivalent pseudomagnetic field in Tesla, and not as a parameter with intrinsic physical meaning.”

If the pseudomagnetic field is to be treated purely as a mathematical parameter without intrinsic physical meaning, then the overall narrative should be revised accordingly. Specifically, the emphasis on the “magnitude” of the pseudomagnetic fields should be removed or substantially reduced. For example, in the conclusion, the authors highlight the novelty of their work by comparing the strength of their pseudomagnetic fields with previous studies. Such claims are misleading if the field magnitude has no real physical significance.

I recommend that the authors instead use the term “exciton splitting” consistently throughout the manuscript including the title to describe their observations, rather than converting it into pseudomagnetic fields. If their claim of a pseudo-Larmor effect is valid, “exciton splitting” alone is sufficient to establish a meaningful connection to the Larmor effect. Presenting results in terms of pseudomagnetic fields, however, risks misleading the community.

2. Charged exciton peak splitting.

It is unclear how the authors extracted the energy splitting from the bottom panel of Fig. 4a. Unlike Fig. 2b, the PL spectrum in Fig. 4a exhibits an additional feature on the lower-energy side, while the lineshape on the higher-energy side remains essentially unchanged. Under these circumstances, it is questionable whether one can reliably deduce the peak splitting that the authors claim.

Reviewer #2

(Remarks to the Author)

First of all, we appreciate the authors' new experiments and clarifications, about which we do not have any questions. However, as we have better understood the authors' method and tried to apply it to the data presented in the manuscript, we have found a significant inconsistency that greatly shakes our confidence about the conclusion of the manuscript. As we explain below, the (coherent) life time reported in the paper cannot reproduce the fit in Fig. 3c, one of the most important results in the paper.

As explained by the authors, the steady-state solution to the Eq. 2 in the manuscript is $S_z = \tau_{\perp} \Omega / [1 + (\tau_{\perp} \Omega)^2]$, where $1/\tau_{\perp} = 1/T_{\text{coh}} + 1/\tau$.

Since the cited exciton lifetime τ is 2 ps, and the fitted coherence time T_{coh} is 1 ps, τ_{\perp} should be 2/3 ps. On the other hand, the solution for S_z is maximized when $\tau_{\perp} \Omega_{\text{max}} = 1$, suggesting $\Omega_{\text{max}} = 1.5$ THz. If we convert this to the energy unit ($\Delta E_x = h \Omega_{\text{max}}$, where h is the Planck constant), it corresponds to a peak splitting $\Delta E_x = 6.2$ meV.

However, such a peak splitting is much larger than any splitting reported by the authors in the main or supplementary. In Fig. S2, the authors showed that in WSe₂ the peak splitting is at most 1 meV under a 0.5% uniaxial strain. Since the largest achievable uniaxial strain is 1.5%, the linearly extrapolated largest splitting at the 25 T limit is only 3 meV, much less than the maximum condition derived above. In other words, the authors shouldn't be able to observe any peak or plateau under their experimental conditions. As a result, we cannot come up with any reasonable fitting that behaves like the one in Fig. 3c.

We consider such an inconsistency significant as it appears also in another part of the analysis. According to our calculation, the maximum ensemble-averaged rotation at 25T is $\tau_{\perp} \Omega = (2/3 \text{ ps}) \cdot (0.73 \text{ THz}) \cdot 2\pi$, which is only $\sim \pi$, instead of 6π as the authors claimed.

We are therefore very concerned that the authors may have overlooked an important factor of 6 (or 2π) throughout their analysis. Such a missing factor has directly undermined their conclusions on the pseudospin rotation physics as well as the coherence time.

Reviewer #3

(Remarks to the Author)

Version 2:

Reviewer comments:

Reviewer #1

(Remarks to the Author)

I would like to thank the authors for their efforts in addressing the issues. While they have partly responded to the concerns, I

remain unconvinced by their argument that providing the effective magnetic field in Tesla is useful for the readers. I respectfully disagree. Even in the references cited in their rebuttal (e.g., Nat. Light: Sci. & Appl. 9, 144 (2020); Nat. Photonics 7, 153 (2013)), the main figures do not explicitly present the real magnetic field values. For instance, in Nat. Photonics 7, 153 (2013), the comparison is made only with energy splitting in graphene under magnetic fields, where a clear physical context exists. By contrast, such context is lacking here. Therefore, I do not think this work is suitable for publication in Nature Communications.

Reviewer #2

(Remarks to the Author)

We appreciate the authors for figuring out their mistake together with ours - the unit of frequency is important but often implied. The difference in the angular and linear frequency, the factor 2π , can lead to wrong conclusions such as reporting half cycle as multiple rounds of rotation. We are glad it is now fixed.

We are also glad to see the authors specify the exciton splitting energy in the new version of the manuscript - it certainly helps the readers to build the physical picture and to compare the phenomena with those they are already familiar with.

We would now support the paper for publication.

Reviewer #3

(Remarks to the Author)

We thank the Reviewers for their thoughtful comments and concerns. To address these issues, we conducted multiple new experiments, re-analyzed the data, and re-wrote/re-focused large parts of the manuscript. The most important changes are:

- We measured the intensity of exciton emission for various angles between the pseudomagnetic field and the initial pseudospin directions. The dependence provides a stronger confirmation for the observation of the pseudo-Larmor effect.
- We conducted two new independent experiments confirming the previously hypothesized magnitude of strain and establishing that no non-linear effects arise at high strain.
- We measured and analysed the pseudo-Zeeman effect via reflectivity. These new results align better with the model of this effect than the photoluminescence data previously reported in the manuscript.
- We rewrote large parts of the manuscript to reflect the main focus of the work better: using pseudomagnetic field to probe, for the first time, the symmetry character of many-body excitonic species.

Overall, we believe that these changes significantly improve our work, making it suitable for *Nature Communications*.

Reviewer #1 (Remarks to the Author):

The manuscript entitled “Fermi polarons under large pseudomagnetic fields” by Denis Yagodkin et al. report observation on luminescence from MoSe₂ and WSe₂ monolayers under strain. The authors claim that pseudospin of excitonic quasi-particles can be manipulated by tunable uniaxial strain which acts like pseudomagnetic fields. Although the claimed observation poses an interesting perspective, I have several concerns and questions on their analysis. Therefore, I cannot recommend the publication in *Nature Communications*.

1. They claim that they can generate pseudomagnetic fields exceeding 40 Tesla. However, it is not clear how this pseudomagnetic field is estimated from observed energy shifts. To quantify pseudomagnetic fields, one would need g -factors of excitonic quasiparticles. In their manuscript, they seemed to use $2\mu_B = 0.116 \text{ meV/T}$ which is attributed to electron's gyromagnetic factor. How does the author justify this g -factor for their observation for neutral excitons and fermi polaron? One should carefully consider g -factors for electrons and holes which can contribute to form excitonic quasiparticles.

Response: We agree with the Reviewer that it is challenging to define the value of the g -factor used to convert the strain-induced splitting to the strength of the pseudomagnetic field. Ultimately, this g -factor is only necessary to develop an intuition about the phenomena that may occur at a specific strength of pseudomagnetic field by connecting it to an equivalent magnetic field. Unfortunately, there is no single accepted method for choosing the g -factor. For an out-of-plane pseudomagnetic field, the most logical way (also followed in the literature, e.g., *Science* 346, 1205 (2014), as the Reviewer pointed out) is to take the g -factor that ensures the excitonic splitting under the pseudomagnetic field equals the splitting under the magnetic field of the same strength. Unfortunately, this approach does not work for the in-plane pseudomagnetic field

studied in the manuscript, since the excitonic g -factor for the real in-plane magnetic field is zero [e.g., *Nat. Comm.* 13, 4997 (2022)].

Therefore, we adopted the free-electron g -factor $g = 2$ as a reasonable quantity free of any microscopic assumptions of the Zeeman effect induced by the real magnetic field. Moreover, $g = 2$ has been used in previous works to describe the electron response to the in-plane magnetic fields in transition-metal dichalcogenide monolayers and to describe the dark-bright exciton mixing, see, e.g., C. Robert, et al. Measurement of the spin-forbidden dark excitons in MoS₂ and MoSe₂ monolayers [*Nat. Comm.*, 11 4037, 2020.]

Alternatively, one can use the bright exciton g -factor for the out-of-plane magnetic field, between ≈ 2 and ≈ 5 depending on the material. This will not change the order-of-magnitude estimates of the field strength.

Nevertheless, we agree that there is a degree of arbitrariness in choosing the “ g -factor”. Because of this and other reasons, in the updated manuscript, we shifted the focus from the strength of the pseudomagnetic field to the phenomena occurring at these fields. In particular, the introductory section was largely rewritten. We especially emphasize the unique capability of in-plane pseudomagnetic field measurements to distinguish between Bosonic and Fermionic many-body states. We also added an explanation of why we consider $g = 2$ the most reasonable value:

We used the free-electron gyromagnetic factor $g=2$ (corresponding to $2\mu_B = 0.116\text{meV/T}$, with μ_B being the Bohr magneton) solely to convert the measured splitting into an equivalent pseudomagnetic field in Tesla, and not as a parameter with intrinsic physical meaning.

2. They show pseudo-Zeeman effect for MoSe₂ monolayers while they show pseudo-Larmor effect for WSe₂ monolayers. Why can they show both effects for the same materials? According to their claim, the pseudo-Zeeman effect is the driving mechanism or pseudo-Larmor effect. For the coherence of the overall discussion, it is much better to choose one material to demonstrate both effects.

Indeed, in the main text, we show the pseudo-Zeeman effect using MoSe₂, while the Larmor-like precession is demonstrated with WSe₂. MoSe₂ was chosen for the pseudo-Zeeman study because its neutral exciton peak stays well-resolved at large strain. Additionally, this material exhibits a simpler excitonic spectrum compared to W-based materials, making quantitative analysis easier. WSe₂ was used for the Larmor study because of the longer excitonic coherence time in this material. Therefore, the use of two different materials is best suited, in our view, to illustrate the physical effects central to our work.

However, we agree with the Reviewer that it is essential to ensure that both effects are observed in a single material. To address this concern, we have now updated Supplementary Figure S2, which shows the pseudo-Zeeman effects in WSe₂. This data complements the pseudo-Larmor effect data in the same material in Figure 3 of the manuscript. Therefore, both effects are observed in WSe₂.

Motivated by the reviewer’s request and for completeness, we also probed for the pseudo-Larmor effect in MoSe₂. The effect is absent even at considerable strain: right- and left-circularly

polarized emission under linearly polarized excitation are equivalent (Fig. R1a-c). We suspect that the reason for this is a short excitonic coherence time in that material. To check this hypothesis, we ran a separate experiment, in which we probed excitonic coherence in a device with circular trench, in which a pseudomagnetic field is absent (Fig. R1d,e). Under moderate strain, the intensity of the photoluminescence co-polarized with the linear excitation is the same as the intensity of orthogonally-polarized emission. This means that excitonic coherence is entirely lost over the exciton's lifetime. For comparison, we observe over $\sim 30\%$ of co-polarized emission in WSe_2 . This is consistent with earlier studies reporting conduction-band splitting close to the LA-phonon energy in MoSe_2 , which drives fast intervalley scattering leading to rapid loss of coherence [PRL 121, 167401, 2018].

We think that these new measurements also clarify one of the Reviewers' concerns: “According to their claim, the pseudo-Zeeman effect is the driving mechanism of pseudo-Larmor effect,” (bold highlights what we believe was a typo, which we corrected). Our measurements show that in the case of MoSe_2 , the pseudo-Zeeman splitting is present. Still, the coherence time is shorter than the exciton's lifetime, which prevents observing the pseudo-Larmor precession. The reason for this is that the Larmor effect is sensitive to shared phase. At the same time, the Zeeman splitting is just a static shift between the energies of the two states and is hence unaffected by the phase coherence.

Figure R1: a-b) Chirality-resolved PL spectra of MoSe_2 under linearly polarized excitation at 10 T and 18 T pseudomagnetic fields show no circular polarization upon linear excitation, indicating a negligible amplitude of pseudo-Larmor effect. **c)** Valley polarization S_z remains near zero across field strengths **d-e)** Co- and cross-polarized PL under strain in circular membrane (to ensure zero pseudomagnetic field) show no polarization, confirming absence of exciton coherence.

3. They claim that pseudospin dynamics can be controlled by strains. They define S_z and $S_{//}$ to describe quantum coherent states of pseudospin in Bloch sphere as shown in their Fig.1c. Under linearly polarized laser excitation, they observe photoluminescence signals that shows stronger circular polarization component for σ^- than σ^+ . They interpretate their observation as a result of pseudo-Larmor effect in Fig.3. However, in order to claim such effects, the authors should systematically show orientation dependence of their linearly polarized excitation with respect to the pseudomagnetic field direction.

When their pseudospin direction is along the pseudomagnetic field direction, they should not observe the circular polarization dependence in their PL.

We thank the reviewer for the insight that may significantly strengthen our work. In fact, we already show that the σ^+/σ^- imbalance, the hallmark of pseudo-Larmor effect, reverses sign when the excitation polarization is rotated by 90° , i.e., when the initial pseudospin is flipped to the opposite side of the Bloch sphere (Supplementary Fig. S6). That can be considered as a subset of experiments suggested by the reviewer. Such a sign reversal demonstrates that the pseudospin rotates in opposite directions for the two orthogonal initial states. This is in full agreement with the expected behavior of the vector product between the pseudospin and pseudomagnetic field vectors that drives the Larmor effect.

We agree with the reviewer that a full angular scan of the excitation polarization would be ideal; however, in most of the optical components used in our setup (and most other conventional optical setups), only the P- and S-polarized components are well preserved, so we cannot vary the incident polarization continuously without losing its purity. Continuously rotating the sample instead of the polarization would solve this problem, but it is not possible in our current setup and is technically challenging to implement.

Figure R2 Pseudo-Larmor effect in WSe_2 at various polarizations of excitation a-b) PL spectra of WSe_2 (distinct from the sample in Fig. 3 of the main text) at 8 T under linear excitation aligned (a) parallel and (b) antiparallel to the pseudomagnetic field show no circular polarization imbalance, as expected when the pseudospin is parallel to Ω . Insets depict excited pseudospin orientation relative to Ω in each case. **c)** When the sample is rotated to create a 45° angle between excitation and Ω , a σ^-/σ^+ asymmetry appears, consistent with pseudo-Larmor precession. The asymmetry is expected to be proportional to the projection of the pseudospin onto the Ω axis, which is maximal at 45° .

Instead, we carried out the suggested measurements for several discrete angles between the pseudomagnetic field and pseudospin. First, we measured changes in pseudospin initially aligned along or opposite to the pseudomagnetic field in WSe₂. To achieve this, we recorded the asymmetry between RCP and LCP emission (pseudospin along the z-axis) for excitation with linearly polarized light along the major (minor) axis of the elliptical TMD membrane. This excitation corresponds to the initial pseudospin along (opposite) to the pseudomagnetic field direction. Despite a high pseudomagnetic field of 8 T, comparable to the field in the main text, we observed no measurable asymmetry between RCP and LCP emission (Fig. R2a,b). Because the pseudomagnetic field and the excited pseudospins are parallel in both cases, the Larmor effect is absent -- in full agreement with the Reviewer's expectations. Meanwhile, after rotating the sample inside the cryostat by about 45-degree (the configuration used in the main text, it corresponds to the excited pseudospin orthogonal to the pseudomagnetic field), we do observe asymmetry between σ^+ and σ^- in emission (Fig. R2c), with the amount of the asymmetry comparable to what is seen in the main text. Again, this is in line with the expectation of the pseudo-Larmor effect.

This new data clarifies the origin of the Larmor effect and is now added to the Supplementary Information as Fig. S7.

4. There are already several works on controlling pseudospin in TMD monolayers using pseudomagnetic fields or real magnetic fields. For example, Nature Materials 14, 290–294 (2015), Science 346, 1205 (2014), Nature Physics 13 (1), 26-29 (2017), and several. They should show how their works provide new novel findings by comparing to the previous works.

While these works are important and relevant, the physics studied here is different. First, the main highlight of our work is the use of a pseudomagnetic field to distinguish between trion and Fermi polaron pictures of charged excitons. To accomplish this, we develop a strain-based approach to a general pseudomagnetic field and characterize its properties. We are not aware of any experimental work that explored this.

Second, the papers mentioned by the Reviewer and that we also cite in our work discuss the optically generated Stark effect. That effect can also be described in terms of a **transient out-of-plane** pseudomagnetic field acting on pseudospin, but with very different properties compared to the strain-related **static in-plane** pseudomagnetic field studied in our work:

- An optically generated pseudomagnetic field is always oriented in the out-of-plane direction, and its effects mimic those of a conventional out-of-plane magnetic field. In contrast, the strain-related pseudomagnetic field in our work is in the in-plane direction, leading to different physical phenomena. For example, its effects do not directly map to those of a conventional magnetic field: the excitonic Zeeman splitting in the in-plane magnetic field is zero.
- The response of excitons to strain-induced (in-plane) pseudomagnetic field allows distinguishing the symmetry properties of excitons (Bosons vs. Fermions) by analysing their pseudomagnetic-field-induced splittings. This central result of our work cannot be achieved with optically induced transient fields, as they break the time-reversal symmetry.

- Optically generated pseudomagnetic field requires ultrastrong laser pulses (field strength is controlled by the fluence) and hence can only be applied transiently while heating the sample. This precludes many potential experiments and makes low-temperature, high-resolution excitonic studies inaccessible. In contrast, strain-induced pseudomagnetic fields are generated statically and do not involve sample heating. As a result, we could study the effect of such a field on “fragile” many-body states. In the future, it will be possible to combine a magnetic out-of-plane field and an in-plane pseudomagnetic field with our approach to gain full control of the valley degree of freedom.

We fully agree that the points above were not clearly explained in the previous version of the manuscript. To clarify this to the reader, we completely rewrote the introduction (the text marked blue is changed) to highlight our focus on the symmetry properties of excitons and the importance of in-plane pseudomagnetic fields. We now explicitly state that we view strain-induced pseudomagnetic fields as a vehicle to explore many-body excitonic states.

Reviewer #2 (Remarks to the Author):

The manuscript titled ‘Fermi polarons under large pseudomagnetic fields’ by Denis Yagodkin et al describes the manipulation of valley pseudospin of neutral excitons in monolayer MoSe₂ and WSe₂ using an uniaxial strain. The authors claim an in-plane pseudomagnetic over 30 T and an enhanced intervalley coherence time which allows up to three full rotations along the polar direction in the Bloch sphere. In the last part, the authors also conclude the nature of the charged exciton, after comparing its pseudo-Zeeman splitting with theoretical predictions. Although the physics discussed in the work is interesting and the theoretical analysis is very thorough, I feel more experimental evidence is needed to support the authors’ claims, as explained in details below.

1. Although the authors claim to have generated a pseudomagnetic field up to 30 T, they only experimentally demonstrate an exciton splitting of 0.7 meV through the polarization dependence of the photoluminescence spectra, which corresponds to a 7-T field. The claimed large field is an extrapolation from the low field limit based on the calibrated strain. I understand the authors’ argument that when the splitting becomes larger, the high energy state is not thermally populated, so the splitting is no longer reflected in PL. But relying on pure theoretical modelling is risky as it contains some assumptions that may not apply to all experimental conditions. For example, under the largest strain condition, the extrapolated exciton splitting becomes comparable to the thermal energy, so the Bose-Einstein distribution should be used, instead of Boltzmann distribution. In the meantime, the uniaxial strain calibration also relies heavily on the finite element analysis in the linear regime, while we know the mechanical problem can become highly nonlinear when the strain is large. Therefore, I would suggest the authors perform some additional experiments to verify some of the key conclusions of their models. For example, I imagine the exciton splitting under large strain can be directly measured in the absorption spectrum through polarization resolved reflection contrast experiments.

We fully agree that the phenomena at large strain/pseudomagnetic field deserve more careful consideration. We therefore conducted several new experiments suggested by the Reviewer.

First, we would like to clarify that the strain values we use in our manuscript were **not** obtained from the finite element analysis. In fact, we used the experimentally measured shift of neutral excitons emission energy as an indicator for the value of the trace of the strain tensor, $\epsilon_{xx} + \epsilon_{yy}$. The relation between these two quantities is well established by the works of our group [*Nat Com* 13, 7691 (2022) and *Nat Com* 15, 7546 (2024)] and that of others [*Nano Research* 14, 1698 (2021)]. To demonstrate this, in Figure R3a, we showed the energy position of excitons vs. strain level determined directly via sensitive optical interferometry, which is completely independent of the PL spectroscopy from which the exciton energy is acquired. In these measurements, we used a device with circular trench to exclude the role of the pseudomagnetic field. We see that the dependence between the excitonic energy position and strain is fully linear within our measurement range. This analysis suggests that the non-linear effects mentioned by the Reviewer are unlikely to affect the trace of the strain tensor $\epsilon_{xx} + \epsilon_{yy}$.

COMSOL modelling was used only to confirm that strain uniaxiality $(\epsilon_{xx} - \epsilon_{yy})/(\epsilon_{xx} + \epsilon_{yy})$ is a geometric quantity that is roughly independent of the trace of the strain tensor. This result can be validated through a simple back-of-the-envelope calculation. We picture two strings of unequal length along the ellipse's major and minor axes, both pulled downward with the same force (Fig. R3b). Simple geometrical estimates shown in Fig. R3b suggest that uniaxiality in an elliptical membrane is roughly given by $(b^2 - a^2)/(a^2 + b^2) \sim 0.75$, where $b=8 \mu\text{m}$ and $a=3 \mu\text{m}$ are the major and minor axes of the elliptical hole supporting the TMD membrane. Note that the depth $d \ll a, b$ is sufficiently small for the whole range of strain. The finite element modelling (COMSOL) gives the uniaxiality of ~ 0.8 , confirming this simple intuition.

Nevertheless, we understand the reviewer's concern given that several observables in our experiments evolve differently at low and high strain. We therefore performed additional polarization-resolved Raman measurements that is a well-established approach to directly probe the uniaxiality of strain. We opted for monolayer MoS_2 , which has clearly separated and well-studied Raman features. We used devices with the same elliptical-trench geometry as in the main text. At zero strain, the E' Raman mode shows no resolvable splitting; with increasing strain, the E' peak demonstrates a clear splitting in energy, a well-established hallmark of uniaxial strain [e.g., *Nature Comm* 8, 1370 (2017), *Phys. Rev. B* 93, 075401 (2016)], while the energy of A_1' mode remains polarization-angle-independent (Fig. R4a-c), also in agreement with previous measurements. From the magnitude of the E' splitting, we extracted the uniaxial strain component $\epsilon_{xx} - \epsilon_{yy}$. Finally, we obtained strain uniaxiality by dividing that component by the total (biaxial) strain obtained from interferometric membrane-deflection measurements (Fig. R4d). We

Figure R3: a) Shift of neutral and charged excitonic peaks in MoSe_2 vs. strain, measured in a circular device to exclude the effects of pseudomagnetic field. **b)** A geometrical estimate of strain uniaxiality in an elliptical membrane.

see that this newly measured uniaxiality is about 0.75 and is roughly strain-independent at large strain within uncertainty. This exactly matches our simple geometric estimates and previous finite-element simulations. Overall, we see that our system is linear even at high loads.

We have incorporated these results and clarifications in the revised manuscript and Supplementary Note S2 (new Raman polarization figure and analysis), which we believe addresses the reviewer's concern regarding the high strain regime.

Second, the concern about the statistics of excitons at high strain is important and valid. In general, the applicability of the Bose-Einstein (compared to the simplified Boltzmann) distribution is determined by the occupancies of excitonic states rather than by the energy splitting. Indeed, the equation for the Bose-Einstein (μ is the chemical potential, T is the temperature and k_B is the Boltzmann constant)

$$f_{BE}(\varepsilon) = \frac{1}{\exp\left(\frac{\varepsilon - \mu}{k_B T}\right) - 1}$$

passes to the Boltzmann one

$$f_B(\varepsilon) = \exp\left(\frac{\mu - \varepsilon}{k_B T}\right)$$

provided that the -1 in the denominator can be neglected, i.e., provided that the exponent $\exp\left(\frac{\varepsilon - \mu}{k_B T}\right)$ is large. It is equivalent to the requirement that the occupancy of a given excitonic state is small, $f_{BE}(\varepsilon) \ll 1$. It is exactly the condition where $f_{BE}(\varepsilon) \approx f_B(\varepsilon) \ll 1$. This condition can be rewritten via the exciton density as the requirement that the exciton density n is sufficiently small

Figure R4: Quantification of uniaxial strain using polarization-resolved Raman spectroscopy. a) Representative Raman spectra of the in-plane E' and out-of-plane A_1 Raman modes under tensile uniaxial strain. The doubly degenerate E' mode splits into E^+ (parallel) and E^- (orthogonal) to the strain axis, while A_1 remains essentially unchanged. **b)** Angular dependence of the E' peak energy versus analyzer angle exhibits the expected four-fold symmetry for an uniaxially split E' mode. **c)** Peak energies of E^+ and E^- versus total (biaxial) strain show linear trends, supporting linearity of uniaxial strain in the entire measurement range. **d)** Uniaxiality U extracted from the E' splitting (small markers) is approximately constant above $\sim 0.5\%$ total strain, when peaks are well resolved. The large markers show a 5-point moving average and the curved line is a guide to the eye. The data agree with the simple geometric estimate $U = (b^2 - a^2)/(b^2 + a^2)$, where $a = 3.3 \mu\text{m}$ and $b = 8.3 \mu\text{m}$ are the axes of the elliptical trench (horizontal line), confirming that the uniaxial-to-biaxial strain ratio is fixed by device geometry and does not degrade at high load.

$$n \ll \frac{\hbar^2}{2mk_B T} \sim 10^{12} \text{ cm}^{-2},$$

where m is the exciton's effective mass.

While the strain itself can cause the exciton redistribution between the split sublevels, the total density of excitons is controlled by the experimental conditions: it is determined by the relation between the exciton generation and decay rates. The fluence used in our experiment corresponds to the exciton density of 10^9 cm^{-2} , three orders of magnitude smaller than the degeneracy density. We therefore believe that it is sufficient, in our case, to use the Boltzmann distribution. To explain this to the reader, we added this argument to the theory part of the Supplementary Information Note S1.2.

Third, we followed the Reviewer's suggestion to directly extract the splitting from the absorption spectroscopy. The main advantage of the approach is that it avoids the main problem of photoluminescence measurements – “dimming” of the higher-lying excitonic state – which leads to the underestimation of the pseudomagnetic field strength. The reason we omitted these measurements previously is that interference effects arising between suspended TMD and Si underneath make the interpretation of such data challenging. Additionally, it makes differential reflectivity types of measurements impossible, as the reference signal does not account for the

Figure R5: Exciton splitting measurements confirming pseudomagnetic fields at high strain. a) The dependence of MoSe₂ reflectivity spectrum on strain. **b)** Polarization-resolved reflectivity at 1.0% strain reveals 2.0 meV splitting, where purple and orange lines correspond to detection polarization set to minor and major axis of the elliptical trench. **c)** Analyzer-angle-resolved reflectivity confirms 2.1 meV pseudo-Zeeman splitting at 26 T. **d)** Extracted exciton splitting versus pseudomagnetic field from PL (blue) and reflectivity (red) measurements; reflectivity better captures the true splitting.

interference in the cavity. For those reasons, such measurement requires a sample with exceptionally strong excitonic resonances and a specific cavity depth.

We now measured a high-quality MoSe₂ sample with an overall exciton shift of about 50 meV, corresponding to ~1.0% uniaxial strain (26 T pseudomagnetic field strength) for which we expect 2.8 meV pseudo-Zeeman splitting (Fig. R5). We performed reflectivity measurements following the strategy of PL measurements reported in the manuscript. More specifically, we acquired reflectivity spectra vs. angle of detection analyzer in the co-polarized scheme (using a half-waveplate right before the objective), which corresponds to the sweep of excitation and detection of pseudospin along the equator of the Bloch sphere. We observed a pseudo-Zeeman splitting of ~2 meV in the excitonic peak, close to what is expected for the pseudomagnetic field, and much larger than the value from PL spectroscopy. That is completely in line with our expectations – in PL spectroscopy, when the pseudospin is aligned opposite to a large pseudomagnetic field, the energy of that state is high, and correspondingly its occupation probability is small at cryogenic temperatures. Effectively, the state “disappears,” leading to the deviation between the measured excitonic shift and the theoretical estimates in the manuscript. The remaining deviation of the measured pseudo-Zeeman shift in reflectivity from the model is likely due to noise and spectra-related complexities during the extraction of the splitting value.

We have now added the results of the reflectivity measurements in Supplementary Information (Fig. S4), and it is mentioned in the discussion of the pseudomagnetic field calibration in the main text to explain this subtle point to the reader:

“At higher strain values, the apparent dependence of pseudomagnetic field becomes sublinear (Fig. S3), which we attribute to a reduced intensity of the higher pseudo-Zeeman-split state when the splitting exceeds the thermal energy ($kBT \approx 1$ meV). The model based on this mechanism closely aligns with the observed behavior of X0 (simulation in Fig. 2c, Note S3) and extracted splitting (dashed line in Fig. 2d). In addition, the splitting is close to the expected value in optical reflectivity measurements (Fig. S4)”

2.If the pseudospin does rotate in the vertical plane for three rounds before it becomes incoherent, I expect the magnetic field dependence of the integration Sz* signal should exhibit some damped oscillation, which is currently missing in Fig. 3c or S2. Actually, there is currently no data for Sz* for pseudomagnetic field larger than 12T despite of the claim...

This is a valid point. “Three full rotations” refers to the *average* precession of the pseudospin *in the ensemble*. Meanwhile, individual excitons decay radiatively fast with a characteristic lifetime of ≈ 2 ps; many of them emit before completing a full cycle. The time-integrated signal Sz* that we measure is thus a mixture of excitons that recombine almost immediately after excitation with zero precession and others that survive longer. This averaging effectively smoothens out any oscillations – seen both in theory and in experiment (see Supplementary Note S1 and Fig. 3). In fact, for the steady-state conditions, the pseudospin z-component can be presented within such a simplistic model as

$$S_z \propto \frac{1}{T_{coh}} \int_0^\infty \sin(\Omega t) e^{-t/T_{coh}} dt = \frac{\Omega T_{coh}}{1 + \Omega^2 T_{coh}^2},$$

where we take into account the probability of the exciton to retain its coherence for the time $t \propto T_{coh}^{-1} e^{-t/T_{coh}}$. While the subintegral expression oscillates, the steady-state value does not show any oscillations as a function of Ω in agreement with Figs. 3 and S2 as well as with the more realistic model presented in the Supplementary Note S1.

To experimentally observe the predicted damped oscillations, we would need a *time-resolved* measurement (e.g. pump-probe Kerr rotation or time-resolved reflectivity). We are developing such an experiment, but implementing pulsed excitation on a suspended, strained membrane is very challenging and definitely outside the scope of this work.

Beyond ≈ 12 T, the valley coherence drops significantly, and the PL intensity of the neutral exciton is also reduced. Under these conditions, the S_z^* signal becomes too weak and noisy to extract reliable values. For this reason, we limited the plots in Fig. 3c and Fig. S8 to fields < 12 T.

To explain this to the reader, we now explicitly mention the reason for the lack of oscillations in the main text:

Intuitively, ensemble averaged S_z probed by PL grows linearly with Ω when the average rotation angle for pseudospins during their lifetime, $\tau_\perp \Omega$, is smaller than 2π . At higher field strengths, the pseudospin undergoes multiple rotations around the Bloch sphere during the exciton lifetime, reducing the average pseudospin polarization similarly to the Hanle effect in real magnetic fields.

and

We also note that the ensemble-averaged rotation for the pseudospins should reach $\tau_\perp \Omega \approx 6\pi$ at the largest induced Ω in WSe_2 , ~ 25 T (Fig. S2)

3. The authors claim the $T_{parallel}$ is enhanced by about ten times and the T_{coh} is doubled as the pseudomagnetic field increases, but I am confused about the experimental evidence. In Fig. S8, the authors made a comparison plot between two theoretical models, one assuming a constant T and the other assuming a field-dependent T . First, in Fig. S8b, my understanding is the data quality cannot distinguish two models. Second, in Fig. S8a, there exists some difference at the large field limit, but the authors' assumption is actually the opposite - the two models should give the same $T_{parallel}$ at the large field limit and different $T_{parallel}$ at the small field limit, where both models somehow fit the data well. Therefore, I do not find sufficient experimental evidence to support the authors' claim.

We thank the reviewer for raising this point. First, our main goal is to validate a well-established theory (Ref. [*Phys. Rev. B* 106, 235313 (2022)]; [*Phys. Rev. B* 89, 201302 (2014)]) which explicitly shows that both the longitudinal (T_{\parallel}) and transverse (T_{\perp}) relaxation times grow with pseudomagnetic field. The physical basis for it is the competition of the strain-induced pseudomagnetic field with the effective fields related to the exciton longitudinal-transversal splitting (caused by the electron-hole exchange interaction), which drives exciton pseudospin relaxation.

It is most obvious in the case of the longitudinal relaxation described by T_{\parallel} : Since the exciton pseudospin precesses in the total field, an increase of the field effectively decreases the pseudospin deviation from the initial direction, resulting in the enhancement of T_{\parallel} . In the quantum language, an increase in the pseudomagnetic field increases the Zeeman splitting and suppresses the spin-flip transitions.

Naturally, the effect of the induced pseudomagnetic field cannot be ignored when the precession due to the pseudomagnetic field within the scattering time becomes significant, i.e., where $\Omega\tau_{sc} > 1$, which in our case occurs when the fields exceed about 5 T. Second, as the reviewer noticed, the fit with the constant T_{\parallel} works well for the S_{\parallel} data, however, it yields a rather large value for $T_{\parallel} \sim 10$ ps. Since at $\Omega = 0$ the T_{\parallel} and T_{\perp} are related, $T_{\parallel} = \frac{1}{2}T_{\perp}$, this value contradicts the results for T_{\perp} . Additionally, this value contradicts literature results for valley decay of excitons in MoSe₂ that shows about 1 ps [Nat. Comm. 13, 4997 (2022)].

Guided by these considerations, we fitted our data with the field-dependent model. We acknowledge, and now state explicitly, that the experimental uncertainties alone cannot distinguish the two models; instead, the decisive evidence is the theoretical constraint combined with independent estimates of the crossover field. Taken together, these arguments leave the field-dependent description as the only physically consistent choice. We added a clarifying sentence to the SI Note S3 and the caption of Fig. S11 to highlight this:

Given that the induced pseudomagnetic precession becomes significant above 5 T (where $\Omega\tau_{coh} > 1$), and that a constant T_{\parallel} model yields unphysical values inconsistent with both theoretical constraints (see Eq. (S6)) and independent experimental results, we adopt the field-dependent model as the only physically consistent framework for describing the data.

and

Note that while the simple model with fixed relaxation times fits the data well, it yields the times inconsistent with results of T_{\perp} , as it violates Eq. (S6): $T_{\parallel}(\Omega = 0) = T_{\perp}(\Omega = 0)/2$.

4. How do the authors perform the doping dependence study of g_p as in Fig. 4d is very unclear to me. For MoSe₂, it seems the sample is initially chemically doped so when no gate voltage is applied, the Fermi level is highest (6 meV) and the strain is minimal. As the gate voltage is applied, the doping becomes smaller and the strain increases. The exciton splitting also increases with pseudomagnetic field, so the uncertainty in g_p decreases as E_f decreases. But if this is the case, why are there some data points near 4 meV which shows negligible strain and significant electrostatic doping? Can the authors provide some doping dependent spectra as in Fig. S2 d? On the other hand, in the WSe₂ sample, why are there some low-strain data points at the end and in the middle of the data series? It doesn't seem consistent with Fig. S2 d either, which suggests the sample is initially intrinsic.

We agree – the extraction of g_p was presented imperfectly. The problem arises from the fact that, in our approach, the gate voltage, in addition to the application of strain, affects the position of the Fermi level inside the sample. To address this challenge, we re-measured the same sample

multiple times. Each measurement run has a different initial doping, and therefore, strain and doping were varied with some degree of independence. Below, we provide a detailed description of the measurement procedure.

In our first measurement run, we record the excitonic PL spectrum vs. gate voltage (e.g., Fig. R6a). From the energy shift of neutral excitons, we extracted the strain (more precisely, the trace of the strain tensor), and from trion-exciton separation, we obtained the Fermi energy. The resulting E_F vs. strain is plotted in Fig. R6b (“Run1”). We see that the sample was initially electron-doped at zero gate voltage. The same can be seen directly from the PL map, where the neutral exciton intensity is lower than that of the charged exciton at zero voltage.

Next, at each strain (and E_F) value, we measured the pseudo-Zeeman effect (exciton energy oscillations vs. polarization) for neutral ($\Delta E_X = \frac{g_p}{2} \hbar\Omega = \hbar\Omega$) and charged excitons ($\Delta E_{X^{+/-}} = \frac{g_p}{2} \hbar\Omega$). We then calculated the g-factor for trions by dividing the pseudo-Zeeman splitting for trions and excitons. These g_p vs. E_F are shown in Fig. R5c (same color as in Fig. R5b for the same run).

We then repeated the same procedure for the same device in three subsequent cool-down/warm-up cycles; each cycle began with a slightly different built-in doping, so every “run” starts at a different Fermi energy (different colours in Fig. R6b correspond to different runs). It is evident that when the strain is, e.g., 1% in each run, the Fermi level is different between the runs. For each run, the extracted $g_p(E_F) = \Delta E_{X^{+/-}}/\Delta E_X$ was added to Fig. R6c with corresponding color coding. Note that the points from different runs, with different doping levels, all fall onto the same parameter-free dependence of $g_p(E_F)$ predicted by theory (solid lines in Fig. R6c). This convinces us that g_p for trions is only determined by the Fermi levels and conforms to theory. We updated Fig. 4d in the main text with Fig. R6c and changed the discussion of g_p to avoid confusion. We also included a condensed version of this argument in the main text.

For WSe_2 , the measurement was run at very low carrier densities; here even a 0.5 meV uncertainty in the exciton-polaron peak propagates into noticeable scatter of the extracted E_F , which explains

Figure R6. Fermi energy dependence of g-factor. **a)** A false color map of MoSe_2 photoluminescence dependence on strain. **b)** Extracted Fermi energy in MoSe_2 vs. strain shows a consistent trend of decreasing E_F for increasing strain within each run, despite high uncertainty. **c)** Exciton splitting versus Fermi energy for MoSe_2 (blue) and WSe_2 (red). Multiple thermal cycles lead to varied initial doping of MoSe_2 hence at the same strain (size of the point) there are different Fermi energies. The solid lines are theoretical predictions. **d)** Zoom in into g-factor dependence of WSe_2 on E_F with each point labeled by its index in the measurement sequence.

the apparently misplaced low-strain points at the beginning of that data series, while there is clear trend of decreasing E_F with increasing strain (see Fig. R6d where we added indexes in the sweep to the data).

We now present the full gate-voltage maps for MoSe_2 as requested by the reviewer (updated Fig. S3) and color-coded in Fig. 4d by run, so the evolution of strain and doping is explicit. We also added a clarifying sentence to the caption of that figure:

“... and color shades mark different experimental runs with different initial carrier densities.”

We hope these additions remove the ambiguities you identified.

5.A technical suggestion - since the excitation condition is critical to the coherent part of PL, the authors may want to clarify that for Fig. 2.

We updated the caption of the figure to better explain the polarization used:

“Polarizations of both excitation and detection are linear and co-polarized”.

Reviewer #3 (Remarks to the Author):

Response to Reviewer 1

1. Definition of the in-plane pseudomagnetic field.

The magnitude of the pseudomagnetic field in their work is essentially a mathematical parameter without a clear connection to any physical property. In their rebuttal, the authors themselves acknowledged the difficulty of physically defining the in-plane excitonic g-factor. As they also cite (Nat. Commun. 13, 499), the in-plane excitonic g-factor is known to be effectively zero. Despite this, they simply apply the free-electron g-factor to excitonic quasiparticles. This approach is inappropriate, since the g-factor of an exciton is determined jointly by both electron and hole contributions.

The authors write in their revised manuscript: “We used the free-electron gyromagnetic factor $g = 2$ solely to convert the measured splitting into an equivalent pseudomagnetic field in Tesla, and not as a parameter with intrinsic physical meaning.”

If the pseudomagnetic field is to be treated purely as a mathematical parameter without intrinsic physical meaning, then the overall narrative should be revised accordingly. Specifically, the emphasis on the “magnitude” of the pseudomagnetic fields should be removed or substantially reduced. For example, in the conclusion, the authors highlight the novelty of their work by comparing the strength of their pseudomagnetic fields with previous studies. Such claims are misleading if the field magnitude has no real physical significance.

I recommend that the authors instead use the term “exciton splitting” consistently throughout the manuscript including the title to describe their observations, rather than converting it into pseudomagnetic fields. If their claim of a pseudo-Larmor effect is valid, “exciton splitting” alone is sufficient to establish a meaningful connection to the Larmor effect. Presenting results in terms of pseudomagnetic fields, however, risks misleading the community.

We appreciate this thoughtful discussion. We followed the suggestion and provided the excitonic splitting value when referring to the pseudomagnetic field in most places. In figures, we have now added a new x-axis in the units of energy, in addition to the previously used units of field. We also removed the claim of “large” pseudomagnetic field and changed the title to “*Fermi polarons under strain-induced pseudomagnetic field*”.

At the same time, we note that using the free-electron gyromagnetic factor to convert energy splitting in various types of synthetic pseudomagnetic fields into magnetic field units is a routinely used practice even for states whose magnetic response differs from that of free electrons. For example, pseudomagnetic fields (in units of Tesla) are reported for photons -- with zero g-factor - - in photonic crystals (e.g., Nat Light: Sci&App **9**, 144 (2020) or Nat Phot **7**, 153 (2013)) or for excitons with in TMDs, that have “normal” magnetic g-factor very different from 2 (Ref. 30 of the manuscript). In all these cases, and in dozens of others, this conversion is convenient because it allows for an intuitive comparison with familiar magnetic phenomena at the same field.

The pseudomagnetic field has intrinsic physical meaning in the sense that it leads to phenomena that are intuitive for a magnetic field analog. For example, the Quantum Hall effect and Landau states observed in the previously mentioned *Nat Light* paper, while a related effect may also appear in our experimental system. In contrast, it is challenging to understand these phenomena from a purely strain or exciton energy splitting perspective if the magnetic field analogy is not made explicit.

Regarding the comment that “*g-factor of an exciton is determined jointly by both electron and hole contributions*” we note that while this is entirely true for the “normal” magnetic *g*-factor, the pseudomagnetic *g*-factor simply establishes the units of pseudomagnetic field for a new degree of freedom.

Because of this, we believe that explaining the extent of the analogy, rather than emphasizing the field magnitude, and providing the field in both energy and field units will minimize the risk of misleading the community, follow well-accepted practices, and provide helpful intuition for the readers. Accordingly, we removed claims of ‘large fields’ from the abstract and the title and revised the introduction to facilitate the debate on the physical relevance of the pseudomagnetic field.

2. Charged exciton peak splitting. It is unclear how the authors extracted the energy splitting from the bottom panel of Fig. 4a. Unlike Fig. 2b, the PL spectrum in Fig. 4a exhibits an additional feature on the lower-energy side, while the lineshape on the higher-energy side remains essentially unchanged. Under these circumstances, it is questionable whether one can reliably deduce the peak splitting that the authors claim.

We apologize for the confusion caused by our unsuccessful attempt to oversimplify the presentation of our data.

The excitonic shifts that the Reviewer was alluding to were obtained not from individual spectra (as the graph we showed may have led one to believe; the tilted background in that spectrum appears as an additional feature), but from **90** spectra measured as a function of the polarization detection angle (Fig. R1). We fit each spectrum with both exciton and trion responses (white line in Fig. R1), since an exciton shift affects the high-energy tail of the trion. We then fit the angular dependence with a sine function and include a linear background term to account for any slight drift. Because each point averages over many spectra, the uncertainty of each splitting value is small. We have updated Figure 4 in the manuscript that led to the confusion, and thank the Reviewer for pointing this out.

Figure R1. Charged exciton Zeeman-like splitting under applied pseudomagnetic field. (a) Photoluminescence (PL) map of MoSe₂ in the region of charged exciton, at the neutral exciton splitting of 2.2 meV (corresponding to 19 T). The white line is the fitted charged exciton peak position. Horizontal lines indicate spectra compared in panels (b) and (c). (b) Comparison of spectra at two extremes (blue vs. red), demonstrating that the apparent asymmetry in earlier plots was due to slight drift at low angles. (c) Same comparison at higher angles (green vs. red), where the splitting is more symmetric.

Response to Reviewer 2

First of all, we appreciate the authors' new experiments and clarifications, about which we do not have any questions. However, as we have better understood the authors' method and tried to apply it to the data presented in the manuscript, we have found a significant inconsistency that greatly shakes our confidence about the conclusion of the manuscript. As we explain below, the (coherent) life time reported in the paper cannot reproduce the fit in Fig. 3c, one of the most important results in the paper. As explained by the authors, the steady-state solution to the Eq. 2 in the manuscript is $S_z = \tau_{\perp} \Omega / [1 + (\tau_{\perp} \Omega)^2]$, where $1/\tau_{\perp} = 1/T_{\text{coh}} + 1/\tau$.

Since the cited exciton lifetime τ is 2 ps, and the fitted coherence time T_{coh} is 1 ps, τ_{\perp} should be 2/3 ps. On the other hand, the solution for S_z is maximized when $\tau_{\perp} \Omega_{\text{max}} = 1$, suggesting $\Omega_{\text{max}} = 1.5$ THz. If we convert this to the energy unit ($\Delta E_x = \hbar \Omega_{\text{max}}$, where \hbar is the Planck constant), it corresponds to a peak splitting $\Delta E_x = 6.2$ meV.

However, such a peak splitting is much larger than any splitting reported by the authors in the main or supplementary. In Fig. S2, the authors showed that in WSe2 the peak splitting is at most 1 meV under a 0.5% uniaxial strain. Since the largest achievable uniaxial strain is 1.5%, the linearly extrapolated largest splitting at the 25 T limit is only 3 meV, much less than the maximum condition derived above. In other words, the authors shouldn't be able to observe any peak or plateau under their experimental conditions. As a result, we cannot come up with any reasonable fitting that behaves like the one in Fig. 3c.

We consider such an inconsistency significant as it appears also in another part of the analysis. According to our calculation, the maximum ensemble-averaged rotation at 25T is $\tau_{\perp} \Omega = (2/3 \text{ ps}) * (0.73 \text{ THz}) * 2\pi$, which is only $\sim \pi$, instead of 6π as the authors claimed.

We are therefore very concerned that the authors may have overlooked an important factor of 6 (or 2π) throughout their analysis. Such a missing factor has directly undermined their conclusions on the pseudospin rotation physics as well as the coherence time.

We thank the Reviewers for their very deep and thoughtful analysis – a great sanity check of our data. We believe that the apparent inconsistency comes from a single typo in the Reviewer's formulas: Ω is an angular frequency (units: rad s^{-1}), not linear frequency (as this analysis implicitly assumes, units of THz). Because of that, when converting frequency to energy, one must use $\Delta E_x = \hbar \Omega_{\text{max}}$, and not $\hbar \Omega_{\text{max}}$.

With these corrected conversions, the exact formulas and numbers used by the Reviewers yield a peak splitting on the order of ~ 1 meV, or 8T in the units of magnetic field. This precisely matches our data in Fig. 3c – which we believe is exactly what the Reviewer expected. Overall, this is a very useful sanity check, which we believe our work passes.

We believe the misunderstanding mostly stems from the unusual form of the formula for the equilibrium z-component of pseudospin under pseudomagnetic field, $S_z = \tau_{\perp} \Omega / [1 + (\tau_{\perp} \Omega)^2]$, which counterintuitively does not contain any factors of 2π . This expression is derived simply from the ensemble-averaging of S_z :

$$S_z \propto T^{-1} \int_0^{\infty} \sin(\Omega t) e^{-\frac{t}{T}} dt = \frac{\Omega T}{1 + \Omega^2 T^2},$$

where the exponential factor describes the decay of the exciton population.

Finally, the Reviewer mentioned a factor of 6π in a later sentence discussing the full rotation angle. This is a mistake on our part – stemming from the confusion with the same formula. We thank the Reviewer for noticing this – that sentence, along with another mention of a factor 2π , is now removed. These sentences do not affect any analysis in the manuscript or its narrative.

Reviewer #1 (Remarks to the Author):

I would like to thank the authors for their efforts in addressing the issues. While they have partly responded to the concerns, I remain unconvinced by their argument that providing the effective magnetic field in Tesla is useful for the readers. I respectfully disagree. Even in the references cited in their rebuttal (e.g., Nat. Light: Sci. & Appl. 9, 144 (2020); Nat. Photonics 7, 153 (2013)), the main figures do not explicitly present the real magnetic field values. For instance, in Nat. Photonics 7, 153 (2013), the comparison is made only with energy splitting in graphene under magnetic fields, where a clear physical context exists. By contrast, such context is lacking here. Therefore, I do not think this work is suitable for publication in Nature Communications.

We thank the Reviewer for the constructive feedback and positive assessment of our previous response. We have already revised the manuscript to report all splitting values primarily in energy units (meV). All figures and captions have been presented with exciton splitting in energy units as the primary quantity. We have also included a note clarifying that any values in Tesla are provided only as a secondary, scale-setting reference:

We used the free-electron gyromagnetic g -factor $g=2$ (corresponding to $2\mu_B=0.116$ meV/T with μ_B being the Bohr magneton) to convert the measured splitting into an equivalent pseudomagnetic field in Tesla solely for easier comparison with conventional magnetic effects. To emphasize the difference between pseudomagnetic and real magnetic fields, we also provide the exciton splitting corresponding to the field in units of energy, whenever appropriate.

We hope this clarifies the reason for reporting the splitting values in units of Tesla as a secondary unit. To further centre the discussion on energy units, we now provide meV values even when discussing pseudomagnetic field calibration.

We appreciate the Reviewer's point regarding prior literature emphasizing energy units. We believe our revision aligns with that literature: energy units are now the default, and Tesla units appear only as an optional, secondary guide to scale. We believe this strikes a balance between clarity for readers who prefer energy units and utility for readers who want to get an intuitive understanding of magnetic-like phenomena that can occur in our system.

We hope these changes address the concern and make the distinction between a pseudomagnetic field and a real magnetic field fully transparent.

Reviewer #2 (Remarks to the Author):

We appreciate the authors for figuring out their mistake together with ours - the unit of frequency is important but often implied. The difference in the angular and linear

frequency, the factor 2π , can lead to wrong conclusions such as reporting half cycle as multiple rounds of rotation. We are glad it is now fixed.

We are also glad to see the authors specify the exciton splitting energy in the new version of the manuscript - it certainly helps the readers to build the physical picture and to compare the phenomena with those they are already familiar with.

We would now support the paper for publication.

Thank you for your questions, corrections, and especially catching the frequency-unit issue, which substantially improved the manuscript. We're grateful for your support of publication.

Reviewer #3 (Remarks to the Author):

Thank you for your thoughtful co-review. We appreciate the time you invested to help us improve the paper.